# Wastewater surveillance of SARS-CoV-2 mutational profiles at a university and its surrounding community reveals a 20G outbreak on campus

**Candice L. Swift** , **Mirza Isanovic, Karlen E. Correa Velez, Sarah C. Sellers, R. Sean Norman** *

Department of Environmental Health Sciences, University of South Carolina, Columbia, SC, United States of America

* rsnorman@sc.edu

## Abstract

Wastewater surveillance of the severe acute respiratory syndrome coronavirus 2 (SARS-CoV-2) has been leveraged during the Coronavirus Disease 2019 (COVID-19) pandemic as a public health tool at the community and building level. In this study, we compare the sequence diversity of SARS-CoV-2 amplified from wastewater influent to the Columbia, South Carolina, metropolitan wastewater treatment plant (WWTP) and the University of South Carolina campus during September 2020, which represents the peak of COVID-19 cases at the university during 2020. A total of 92 unique mutations were detected across all WWTP influent and campus samples, with the highest frequency mutations corresponding to the SARS-CoV-2 20C and 20G clades. Signature mutations for the 20G clade dominated SARS-CoV-2 sequences amplified from localized wastewater samples collected at the University of South Carolina, suggesting that the peak in COVID-19 cases during early September 2020 was caused by an outbreak of the 20G lineage. Thirteen mutations were shared between the university building-level wastewater samples and the WWTP influent collected in September 2020, 62% of which were nonsynonymous substitutions. Co-occurrence of mutations was used as a similarity metric to compare wastewater samples. Three pairs of mutations co-occurred in university wastewater and WWTP influent during September 2020. Thirty percent of the detected mutations, including 12 pairs of concurrent mutations, were only detected in university samples. This report affirms the close relationship between the prevalent SARS-CoV-2 genotypes of the student population at a university campus and those of the surrounding community. However, this study also suggests that wastewater surveillance at the building-level at a university offers important insight by capturing sequence diversity that was not apparent in the WWTP influent, thus offering a balance between the community-level wastewater and clinical sequencing.

**Data Availability Statement:** S1-S4 Datasets are available through Mendeley Data at: https://data.mendeley.com/datasets/zz3k2gjg94/1. S1 Dataset

contains the one-way analysis of variance (ANOVA) and subsequent Tukey's test comparing 20G mutation read frequencies between UofSC, WWTP influent collected in September 2020 (treatment "fall"), and WWTP influent collected in July/August 2020 (treatment "summer"). S2-S4 Datasets contain the ARTIC minion (nanopolish) VCF output for Columbia WWTP influent samples from July and August 2020 (S2 Dataset), Columbia WWTP influent samples from September 2020 (S3 Dataset), or University of South Carolina samples from September 2020 (S4 Dataset). Single nucleotide variant analysis is included in the *.vcf files in the output subfolder. Please refer to https://artic.readthedocs.io/en/latest/ for details of the artic minion output. Sequencing reads aligned to the SARS-CoV-2 genome (accession MN908947.3) in BAM format are available at NCBI BioProject PRJNA763484.

**Funding:** Funding sources to RSN: Centers for Disease Control and Prevention (https://www.cdc.gov/) #75D-301-18C-02903, South Carolina Department of Health and Environmental Control (https://scdhec.gov/) #EQ-0-654, and the University of South Carolina. The funders had no role in study design, data collection and analysis, decision to publish, or preparation of the manuscript.

**Competing interests:** The authors have declared that no competing interests exist.

## Introduction

Severe acute respiratory syndrome coronavirus 2 (SARS-CoV-2) is the causative agent for the Coronavirus Disease 2019 (COVID-19) pandemic. Since SARS-CoV-2 ribonucleic acid (RNA) was first detected in the feces of infected individuals [1], its presence has been confirmed in the wastewater of many countries [2–4]. SARS-CoV-2 wastewater surveillance offers many benefits, including, but not limited to, early detection [5], the ability to monitor infection trends separately from clinical data [6], and data that is independent of healthcare access or choices [6]. Collaborations between wastewater surveillance research teams and policymakers have resulted in effective public health actions [7].

Students at universities are at risk for SARS-CoV-2 infection due to factors such as living in high-density facilities (dormitories) in close contact with others. A study of 16,101 university students from Fall 2020 to Spring 2021 demonstrated that although 84% of student were protected from SARS-CoV-2 infection, 16% of students remained susceptible to infection and reinfection occurred in 2.2% of the previously infected student population during the period 12 to 30 weeks after initial infection. [8], prior to the availability of vaccines and the emergence of the highly transmissible delta variant [9] of SARS-CoV-2. Localized wastewater sampling at universities has been used across the United States as a disease mitigation strategy [10–12]. Since wastewater trends can precede clinical data by as much as a week [7], administrative officials can take action quickly. Disease mitigation strategies can include increased COVID-19 testing at specific buildings, also called surge testing [12]. Monitoring at the building level on a college campus has been reported as a highly sensitive method capable of detecting a single asymptomatic student amidst 150–200 individuals [12].

The University of South Carolina serves 27,502 undergraduate students during the academic year [13], representing 21% of the population of Columbia, South Carolina (131,674 as of July 2021 [14]), or 8% of the greater Columbia metropolitan area served by the Columbia wastewater treatment plant. Although it is anticipated that the influx of students at the start of the fall semester would increase transmission of SARS-CoV-2, both due to the increase in population as well as the input of potentially more infectious viral genotypes from other states and countries, the impact of the student population on the community in terms of the SARS-CoV-2 sequence diversity has not been shown in wastewater data. However, the increase in transmission due to the influx of students has been demonstrated by clinical data [15, 16] and predicted by modelling [17] in other communities. We hypothesized that there would be substantial overlap in the observed mutations between wastewater samples collected from the university and those collected from the neighboring WWTP influent, which serves the greater Columbia metropolitan area and the university.

In this work, we compare the sequence diversity of SARS-CoV-2 in wastewater collected from four sites across the University of South Carolina (UofSC) campus in September 2020 and the influent to the Columbia wastewater treatment plant (WWTP), which serves approximately 363,714 individuals in Columbia, SC (based on the United States 2020 Census tabulated by ZIP code), including the UofSC campus, from July-September 2020. The University of South Carolina partially resumed in-person instruction on August 20, 2020. COVID-19 cases of isolation (individuals who tested positive for COVID-19) or quarantine (individuals with close contact to a confirmed case of COVID-19) for the university peaked at 258 from August 30 to September 1 (Fig 1). The results presented here imply that disease mitigation strategies adopted by a university can impact the community at large.

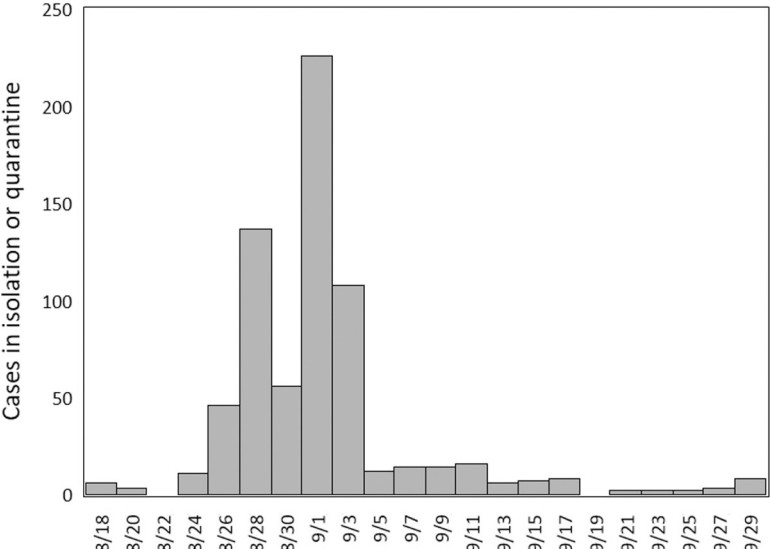

**Fig 1. Histogram of COVID-19 cases of exposure requiring quarantine or isolation at the start of the 2020 academic year.** Each date represents the inclusive end date of a two-day interval.

## Methods and materials

### Wastewater sampling

The Columbia WWTP is a secondary (activated sludge) WWTP that treats municipal wastewater from a population of 363,714 (based on the United States 2020 Census data tabulated by ZIP code [18]) with 6% of total flow permitted from industry. The monthly average flow of the Columbia WWTP is 45 million gallons per day (MGD). One liter 24-hour composite wastewater samples were collected using an ISCO refrigerated autosampler (Lincoln, NE) twice a week at the influent site of the Columbia WWTP. Samples for the University of South Carolina buildings were 0.3 L grab samples of raw wastewater collected between 8:30 and 10:00 AM from the sites marked on Fig 2.

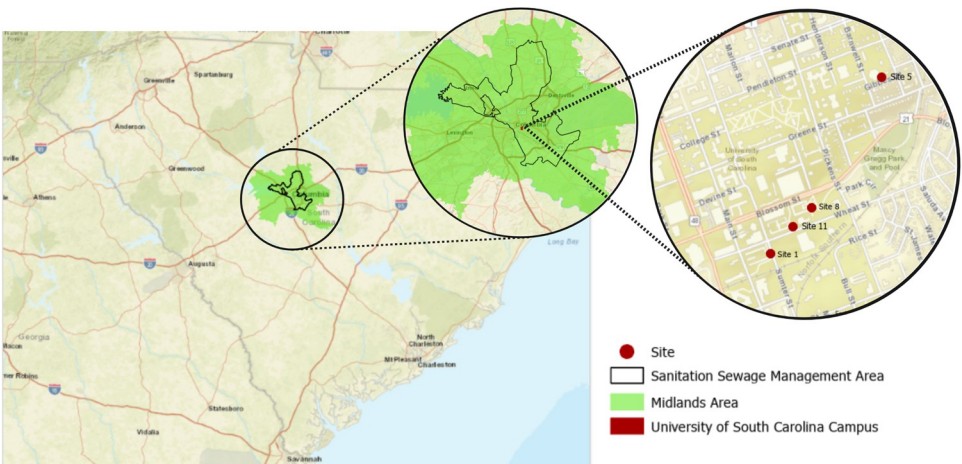

**Fig 2. Locations of wastewater sampling at the University of South Carolina within the greater Columbia metropolitan area.** Figure was rendered using ArcGIS Pro. Green area represents the ZIP codes within the greater Columbia metropolitan area. Black outlined region represents the sanitation sewage management area. Red dots represent the sampling sites within the University of South Carolina campus. Numbered sites not included in this study are not shown.

Columbia WWTP influent samples and university samples were processed separately. One mL of bovine respiratory syncytial virus (BRSV) vaccine (~80 million copies/mL) (INFORCE 3®) was added to one liter of wastewater prior to concentration in order to quantify processing and viral extraction efficiency. The average BRSV viral recovery was 4–5% for WWTP influent samples and 5–8% for university samples. Both influent and university samples were homogenized for 10 min using laboratory blenders. 50 mL (university samples) or 250 mL (WWTP influent samples) of homogenized wastewater was decanted into VWR 50 mL Falcon tubes (university samples) or centrifuge bottles (WWTP influent samples) and were centrifuged using an Avanti® J-E Centrifuge (Beckman Coulter Lifesciences, Indianapolis, Indiana) with a JS-5.3 rotor for 20 min (university samples) or 30 min (WWTP influent samples) at 4,577 $g$ without braking. 50 mL of each supernatant was concentrated to 400 µL using Milipore Amicon 30 kDa ultrafilters (Burlington, MA).

## RNA extraction, library preparation and sequencing

RNA was extracted from 200 µL of the concentrated supernatant using the Qiagen AllPrep PowerViral extraction kit (Hilden, Germany) per the manufacturer's instructions, eluted into 51 µL of RNase-free water, and stored at -80 ˚C until library preparation. Sequencing libraries were prepared following the Oxford Nanopore Technologies (ONT) PCR tiling of SARS-CoV-2 with native barcoding protocol, which is based on the protocol developed by the ARTIC network [19]. The ONT Native Barcoding Expansion 96 (EXP-NBD196) was used. Samples were separated into two different library preparations: July/August 2020 samples and September 2020 samples. Total RNA was transcribed into cDNA using the LunaScript® RT SuperMix Kit (New England Biolabs, Ipswich, MA). The resulting products were amplified by 40 cycles of PCR using two different primer pools (V3 design) to create ~400 bp amplicons spanning the entire SARS-CoV-2 genome. The PCR products were pooled and purified using a 1:1 ratio of SPRISelect beads (Beckman Coulter Lifesciences, Indianapolis, IN). The PCR products were then end-prepped using the NEBnext® Ultra™ II End Repair/dA-Tailing Module (New England Biolabs, Ipswich, MA). Sequencing barcodes and adapters (Oxford Nanopore Technologies, Oxford, UK) were sequentially ligated, and all remaining bead cleanups were performed using SPRIselect beads (Beckman Coulter Lifesciences, Indianapolis, IN). The final libraries were loaded onto two separate R9.4.1 flow cells (Oxford Nanopore Technologies, Oxford, UK) and sequenced using a GridION X5. Columbia WWTP influent samples and University of South Carolina campus wastewater samples from September 2020 (S1 and S2 Tables) were sequenced together on an R9.4.1 flow cell. 7.3 M reads (3.9 Gb) were sequenced with a mean read quality score of 11.1 and a mean read length of 531.6 bp. Columbia WWTP influent samples from July and August 2020 (S1 Table) were sequenced on a separate R9.4.1 flow cell. 3.6 M reads (1.7 Gb) were sequenced with a mean read quality score of 11.2 and a mean read length of 477 bp.

## Data processing

Sequencing data processing was performed according to the ARTIC network "nCoV-2019 novel coronavirus bioinformatics protocol" [20]. Basecalling and demultiplexing were performed within MinKNOW using the high-accuracy model of Guppy version 4.2.3 developed by Oxford Nanopore Technologies. The minimum barcode score was set to 40 and the dual barcoding option was applied. Reads were filtered using a Qscore threshold of 7 and reads outside of the length range of 400–700 bp were omitted to eliminate chimeric reads. Lastly, filtered reads were mapped to the SARS-CoV-2 genome (accession MN908947.3) using minimap [21] within the artic minion command with the V3 primer scheme, filtered aggregate FASTQ file,

and FAST5 directory as input and the normalization option enabled (—normalize 200). Variant calls made by nanopolish were also output from the artic minion pipeline for positions with at least 20× sequencing depth. Mutations identified within primer-binding regions were not considered.

## Principal component analysis (PCA) and analysis of variance (ANOVA)

Principal component analysis (PCA) was conducted following the precedent established by Fontenele and colleagues [22]. Briefly, a genotype for each sample was established by recording the nucleotide frequency at each position in the SARS-CoV-2 reference genome using the Python utility pysamstats. Similarity indices between all pairwise combinations of samples were calculated per Yue and Clayton [23]. The sum of the indices across all positions for all sample pairs was used to construct a distance matrix. The R [24] package prcomp was used to construct a PCA object that was subsequently visualized with the package ggbiplots. One-way analysis of variance (ANOVA) and Tukey's test were conducted in R using the aov() and TukeyHSD() functions, respectively.

## Co-occurrence analysis

Mutational co-occurrence was calculated using the R package cooccur [25], which is a probabilistic model originally developed to analyze species co-occurrence in ecology, but which is broadly applicable to detect statistically significant co-occurrence patterns in other fields [26]. The input data frame to the cooccur function consisted of rows representing the presence or absence (one or zero as values) of all mutations detected by nanopolish at positions with greater than 20× sequencing depth with each wastewater sample represented by a column. Only wastewater samples with at least 50% SARS-CoV-2 genome coverage were included (see S1 and S2 Tables). Concurrent mutations were validated against clinical data using the Global Evaluation of SARS-CoV-2/nCoV-19 Sequences (GESS) database [27] to infer whether the mutations might have co-occurred in the same genome.

## Results and discussion

To gain insight into the influence of a college campus on the surrounding community during the COVID-19 pandemic, SARS-CoV-2 was amplified from both university building-level wastewater and the Columbia metropolitan wastewater treatment plant (WWTP) influent. The WWTP influent was sampled from July to September 2020. University wastewater surveillance began on August 14, 2020, during the week preceding the academic term. Mutations from the SARS-CoV-2 reference genome (accession MN908947.3) and their relative positions in the SARS-CoV-2 genome that were detected in both university wastewater and WWTP influent are depicted in Fig 3, with a table of the corresponding amino acid substitutions for the nonsynonymous mutations.

### Principal component analysis of SARS-CoV-2 sequence diversity in Columbia wastewater treatment plant influent shows little alternation from July to September 2020

SARS-CoV-2 genomic diversity in wastewater was visualized by principal component analysis (Fig 4) using a method pioneered by Fontenele and colleagues [22], in which the sum of Yue and Clayton similarity indices [23] across the entire SARS-CoV-2 genome for all pairwise combinations of samples is the input matrix for the PCA (see Methods) [23]. Notably, data points are overlapping for the University of South Carolina site 1 (Figs 2 and 4B) on August 28

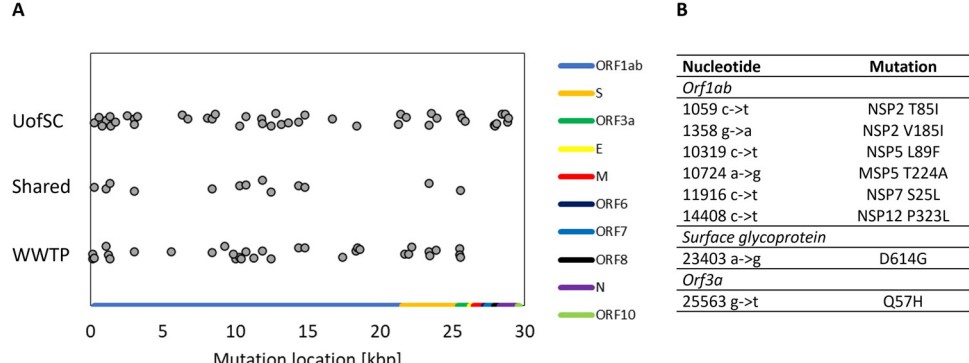

**Fig 3. Mutational profiles of SARS-CoV-2 amplified from University of South Carolina (UofSC) building wastewater and Columbia metropolitan wastewater treatment plant (WWTP) influent.** (A) Location of mutations detected in university wastewater samples and Columbia WWTP influent during September 2020. Dot Jitter was added for clarity to arbitrary y-axis values. (B) Substitutions identified in both university and Columbia WWTP influent. Only nonsynonymous mutations located in coding regions are shown in the table.

and September 11. However, these data points are resolved in a three-dimensional PCA (S1 Fig). One of the limitations of the method pioneered by Fontenele and colleagues [22] is that differences in sequencing depth between samples will affect the sum of the similarity indices. Due to the differences in the sequencing depth of Columbia WWTP influent samples compared to the University of South Carolina samples (S1 and S2 Tables), the sum of the Yue and Clayton similarity indices was higher for pairwise combinations with greater sequencing depth (e.g. two University of South Carolina samples). Therefore, PCA was only conducted for samples of similar depth (Fig 4 panels A and B). PCA for WWTP influent samples (Fig 4A) showed a high degree of similarity between samples collected during the summer (July and August) preceding the start of the academic year and those collected in September 2020. Therefore, despite the return of some students to campus, there was not a substantial shift in the SARS-CoV-2 sequence diversity. Some factors that may contribute to this observation include the continued presence of students on campus during the summer as well as the fact that some

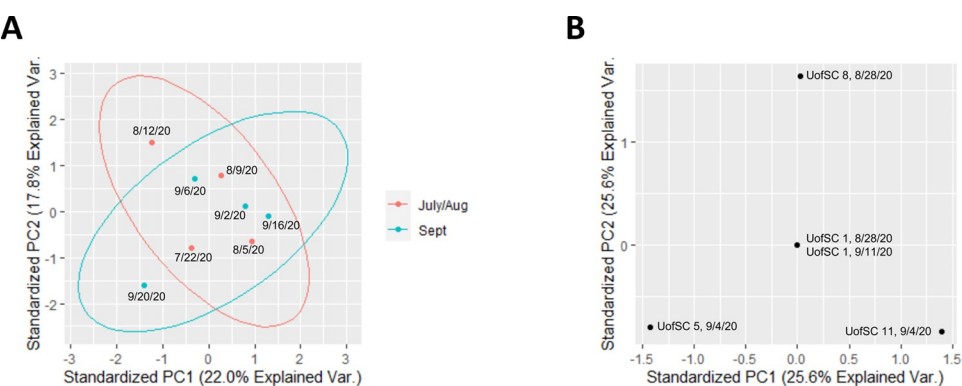

**Fig 4. Principal Component Analysis (PCA) of (A) Columbia WWTP influent composite genotypes (from July-September, and (B) University of South Carolina wastewater composite genotypes.** Composite genotypes for each wastewater sample were established by calculating the nucleotide frequency at each position in the SARS-CoV-2 reference genome. The composite genotypes were then pairwise compared to each other by summing of the Yue and Clayton similarity index [23] for each position in the reference genome. Analysis and visualization were performed with the R packages prcomp and ggbiplots. The size of the ellipse in Normal probability (ellipse.prob option) was set to 0.95.

students continued remote instruction during fall 2020. Also, since the University of South Carolina is a public school, ~60% of its students are from South Carolina [28], which lessens the influx of SARS-CoV-2 genotypes from other states and countries. We anticipate that private colleges and universities in which a greater proportion of the students live out-of-state may experience a shift in SARS-CoV-2 sequence diversity at the start of the academic year. However, it must also be considered that contributions to WWTP influent from industry or stormwater may dilute out the anticipated effects.

## Hierarchical clustering of mutation read frequencies reveals a localized outbreak of the 20G clade at the University of South Carolina

A heatmap with hierarchical clustering of the read frequency (the fraction of reads containing the mutated nucleotide at a specific location) of mutations in the SARS-CoV-2 genome from all wastewater samples (UofSC and Columbia WWTP influent) is depicted in Fig 5, with labeled mutations available in S2 Fig. An intrinsic challenge and limitation of this study was the high degree of biological variability within each condition. The data used for this study was part of a statewide and university sampling effort in response to the COVID-19 pandemic. Samples varied both in time and space for the university samples and in time for the WWTP influent samples. Despite these limitations, the university samples from August 28 (sites 1 and 8, Fig 2) and September 4 (site 5, Fig 2) showed a high degree of reproducibility, as did the mutational profiles of WWTP influent samples collected on September 2 and 6 (Fig 5).

Hierarchical clustering analysis revealed four clusters of mutations (Fig 5). The largest cluster (Cluster 2) consisted of mutations that were mostly detected in a single sample. In contrast, the majority of the mutations in the top cluster of Fig 5 (Cluster 3) were shared across all groups (WWTP influent collected during summer months and September and UofSC wastewater collected during September). Four of the five signature mutations of the 20C clade in NextStrain [29], also referred to as the GH clade in GISAID [30]. were observed in Cluster 3:

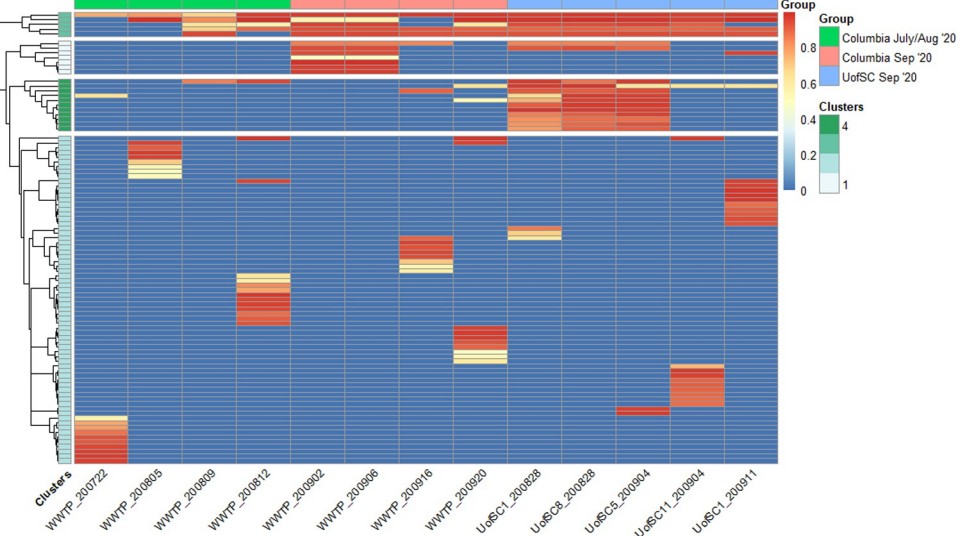

**Fig 5. Read frequency of SARS-CoV-2 mutations in Columbia WWTP influent and the University of South Carolina.** Each cell represents the read frequency of a mutation from the SARS-CoV-2 reference genome (accession MN908947.3) in wastewater samples from the Columbia WWTP influent ("Columbia") or the University of South Carolina campus ("UofSC"). Cells in blue indicate the mutation was not observed in the sample. See S2 Fig for complete list of nucleotide mutations corresponding to the heatmap. Only wastewater samples with at least 50% SARS-CoV-2 genome coverage were included in the heatmap.

241c > t, 1059c > t, 3037c > t, 14,408c > t, and 25,563g > t. The high frequency of these mutations across both university and WWTP influent samples is in agreement with clinical data from July to September 2020, where the 20A, 20C, and 20G NextStrain clades were dominant in the United States [29].

Notably, Cluster 4 of Fig 5 included five of the seven signature mutations of the 20G clade, illustrating that the peak in COVID-19 cases experienced at the beginning of September may have been the result of a 20G outbreak on campus. Cluster 1 indicated traces of the 20G clade in the WWTP influent (10,319c > t). However, the 20G clade was predominantly detected in the localized wastewater sample at the University of South Carolina. The remaining mutations in Cluster 1 demonstrated homogeneity in the WWTP influent collected in early September.

One-way analysis of variance (ANOVA) and subsequent Tukey's test for each of the observed 20G mutations resulted in statistically significant differences (p-adjusted < 0.1) between the University of South Carolina September wastewater samples and WWTP influent collected September (S1 Dataset) for mutations 27,964c > t, 28,472c > t, 28,869c > t, and 25,907g > t. Observation of the mutational profiles of each wastewater sample indicated that the 20G outbreak may have been limited to specific buildings on campus, since the mutational profile for wastewater collected from site 11 differed from site 5 on September 4th (Figs 2 and 5, S2 Fig). These results suggest that localized sampling increases the sensitivity of detection of specific mutations, since in all cases the read frequency of the mutation was higher in the university sample set, whereas the mutations were either not detected consistently or were detected at low frequencies in the WWTP influent sample sets. Nevertheless, it cannot be ruled out that differences in sequencing depth between the university samples and WWTP influent samples, likely caused by the different input RNA concentrations (S1 and S2 Tables) contributed to these perceived differences.

**Co-occurrence of mutations corroborates a 20G viral outbreak at the University of South Carolina.** Mutations that occur in the same SARS-CoV-2 genome can have important phenotypic implications, such as greater infectivity, as demonstrated for the delta SARS-CoV-2 variant [9]. In the context of SARS-CoV-2 amplicons from wastewater, it is difficult to determine which of the mutations originates from the same genome, since the wastewater sample represents a composite from many individuals. However, concurrent mutations can be compared across wastewater samples as a similarity metric and further validated with clinical data to determine whether they are commonly observed in the same viral genome (S3 Table).

Out of 92 total distinct mutations detected in the SARS-CoV-2 genome in the University of South Carolina and Columbia WWTP influent samples, 16 pairs co-occurred more than expected if the two mutations were distributed randomly from each other, as determined using a probabilistic co-occurrence model [26] (Fig 6). In this model, combinatorics is used to compare the observed co-occurrence to the mathematical expected co-occurrence (the product of each mutation's probability of occurrence multiplied by the number of samples). If the frequency of co-occurrence is observed significantly more than expected, then the mutations are considered positively correlated. For the full details of the model, the reader is referred to references [25, 26]. Out of 16 total concurrent pairs of mutations, 12 had at least one signature mutation of the 20G lineage and one had a signature mutation (14,408c > t) of the 20C lineage. Network analysis of concurrent mutations suggests that localized sampling may be more sensitive to detect viral strains, since 12 of the concurrent mutation pairs that were identified in the university samples from September 2020 were not detected in the WWTP influent. However, these results also corroborate a 20G outbreak at the University of South Carolina during September 2020 since 10 of 12 mutation pairs comprised at least one signature mutation of the 20G clade. All concurrent mutations were identified in clinical sequences with a non-zero

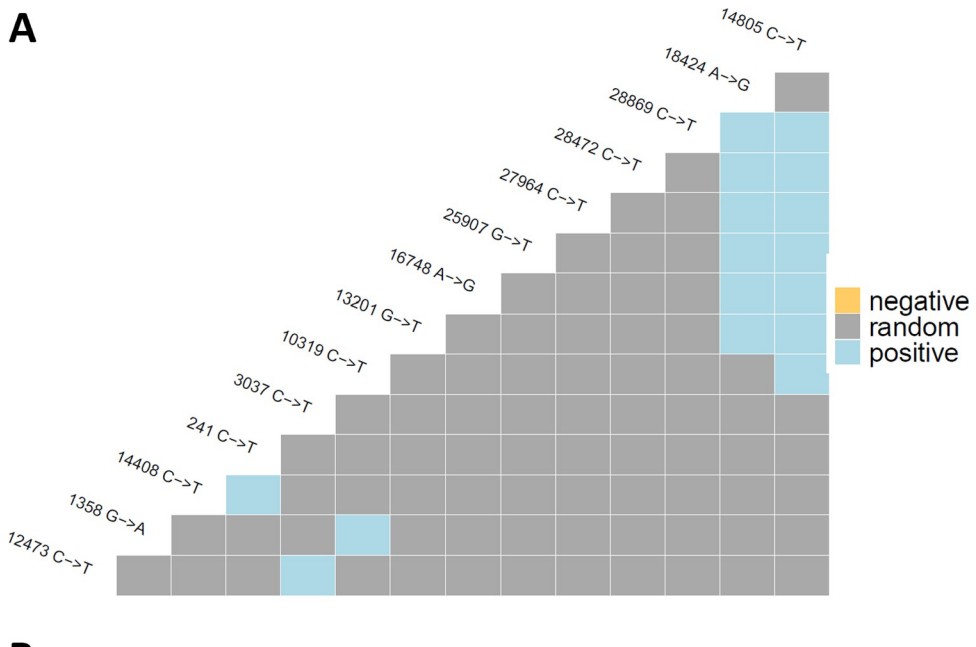

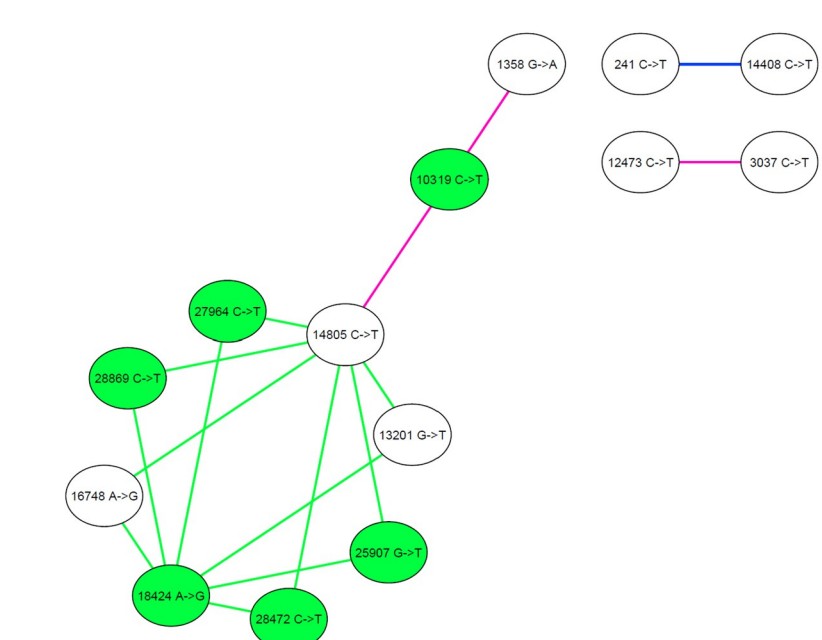

**Fig 6. Co-occurrence and network analysis of mutations detected in WWTP influent and university samples.** (A) Co-occurrence of mutations. No negative co-occurrences (those mutations detected together less often than expected by chance) were identified. (B) Co-occurrence network for mutations from the SARS-CoV-2 reference genome (accession MN908947.3) detected in University of South Carolina wastewater and Columbia WWTP influent. Network was rendered in Cytoscape [31] using the yFiles circular layout algorithm. Edge colors signify the sample types and collection period where the co-occurrences were detected: Green = co-occurrence found in university wastewater samples from September 2020, blue = co-occurrence found in all wastewater sample types (WWTP influent from July/August 2020, and WWTP influent and university samples from September 2020), pink = co-occurrence found in WWTP influent from September 2020 and university samples. Green nodes indicate signature mutations of clade 20G.

concurrence ratio (S3 Table). Therefore, it cannot be ruled out that each pair of concurrent mutations may have originated from the same genome.

Taken together, these results suggest that although localized university wastewater sampling shared concurrent mutations that were also detected in WWTP influent, the greater sensitivity afforded by onsite sample collection nearer the source resulted in the detection of a distinct set of mutations and a strong signal of the 20G clade.

## Conclusions

Although many publications have compared clinical sequence data to wastewater data [22, 32–34], this work represents one of the few studies to compare wastewater data collected from localized sampling at a university to WWTP influent from the greater metropolitan area. This work affirms a close relationship between SARS-CoV-2 sequences from the student body of a university and those of the greater surrounding metropolitan area. Thirteen mutations were identified in both university and WWTP influent samples during September 2020. In addition, we found ten concurrent mutations unique to the localized university sampling that were strongly indicative of a 20G outbreak on campus. Therefore, strategic localized sampling at potential hotspots offers distinctive advantages compared to WWTP influent sampling, such as increased sensitivity in detecting SARS-CoV-2 variants. Relative to sequencing clinical samples or WWTP influent, sequencing at the building level affords a balance between sensitivity and cost.

We anticipate that similar results would be obtained for other universities and their surrounding communities, with even more overlap in cases where universities are situated in less populated areas. Given the overlap in viral mutations between the greater Columbia metropolitan wastewater and the localized university wastewater, university policy makers should work together with government officials from the surroundings communities to manage infectious disease spread.

## Supporting information

**S1 Text. Reverse transcription quantitative PCR (RT-qPCR) methods.**
(DOCX)

**S1 Table. Columbia WWTP influent samples used in this study and sequencing depth and coverage per barcode.** WW = wastewater. Samples with less than 50% coverage that are highlighted in gray were not included in the heatmap or co-occurrence analysis.
(DOCX)

**S2 Table. University of South Carolina campus samples used in this study and sequencing depth and coverage per barcode.** WW = wastewater; UofSC = University of South Carolina.
(DOCX)

**S3 Table. Concurrent mutations identified in wastewater samples that were validated with GESS [27] on December 22, 2021.**
(DOCX)

**S1 Fig. Three-dimensional principal component analysis (PCA) visualized using the pca3d () function in R corresponding to Fig 4B in the main text.**
(TIF)

**S2 Fig. Heatmap corresponding to Fig 1 in the main text in PDF format to enhance visualization of nucleotide mutations.**
(PDF)

## Acknowledgments

We are grateful to the South Carolina utilities directors and operators, as well as SCDHEC, for their contributions to wastewater sampling and transportation. We acknowledge the University of South Carolina facilities staff, as well as Emily Gosnell, Dillon Bryant, Stefano Belmonte, and Sejla Isanovic, for their contribution to the wastewater collection at the University of South Carolina. We would like to acknowledge that the Research Computing program under the Division of Information Technology at the University of South Carolina contributed to the results in this research by providing High Performance Computing resources and expertise. We are very grateful to GISAID Initiative and all its data contributors, i.e. the Authors from the Originating laboratories responsible for obtaining the specimens and the Submitting laboratories where genetic sequence data were generated and shared via the GISAID Initiative, on which this research is based.

## Author Contributions

**Conceptualization:** R. Sean Norman.

**Formal analysis:** Candice L. Swift.

**Funding acquisition:** R. Sean Norman.

**Investigation:** Candice L. Swift, Mirza Isanovic, Karlen E. Correa Velez, Sarah C. Sellers.

**Supervision:** R. Sean Norman.

**Writing – original draft:** Candice L. Swift.

**Writing – review & editing:** Mirza Isanovic, Karlen E. Correa Velez, Sarah C. Sellers, R. Sean Norman.

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
