## [Decision Letter · Decision Letter 0]

13 Dec 2021

PONE-D-21-32875Wastewater surveillance illustrates overlapping SARS-CoV-2 mutational profiles between a university campus and its surrounding communityPLOS ONE

Dear Dr. Norman,

Thank you for submitting your manuscript to PLOS ONE. After careful consideration, we feel that it has merit but does not fully meet PLOS ONE’s publication criteria as it currently stands. Therefore, we invite you to submit a revised version of the manuscript that addresses the points raised during the review process.

The two reviewers of your manuscript provided extensive and detailed comments and requests for improvements or clarifications. Please read through both reviews carefully and address each of the points made by the reviewers as thoroughly as possible. In some cases additional analyses are requested, such as ANOVA. 

We look forward to receiving your revised manuscript.

Kind regards,

Theodore Raymond Muth

Academic Editor

PLOS ONE

3. We note that Figure 2 in your submission contain map images which may be copyrighted. All PLOS content is published under the Creative Commons Attribution License (CC BY 4.0), which means that the manuscript, images, and Supporting Information files will be freely available online, and any third party is permitted to access, download, copy, distribute, and use these materials in any way, even commercially, with proper attribution. For these reasons, we cannot publish previously copyrighted maps or satellite images created using proprietary data, such as Google software (Google Maps, Street View, and Earth). For more information, see our copyright guidelines: http://journals.plos.org/plosone/s/licenses-and-copyright.

Reviewers' comments:

Reviewer's Responses to Questions

**Comments to the Author**

1. Is the manuscript technically sound, and do the data support the conclusions?

Reviewer #1: Partly

Reviewer #2: Partly

2. Has the statistical analysis been performed appropriately and rigorously? 

Reviewer #1: No

Reviewer #2: Yes

3. Have the authors made all data underlying the findings in their manuscript fully available?

Reviewer #1: Yes

Reviewer #2: Yes

4. Is the manuscript presented in an intelligible fashion and written in standard English?

Reviewer #1: Yes

Reviewer #2: Yes

5. Review Comments to the Author

Reviewer #1: The authors report and compare the sequence diversity of SARS-CoV-2 from wastewater influent to the Columbia, South Carolina, metropolitan wastewater treatment plant (WWTP) and four sites from the University of South Carolina (UofSC) campus during July-September 2020. In this study, 154 unique mutations were detected across all samples; of those 26 mutations were shared between the university and the WWTP. Their main conclusions are: 1- there is a close relationship between the prevalent SARS-CoV-2 genotypes from the UofSC campus and the Columbia WWTP.

2- wastewater surveillance at the building level at a university captures sequence diversity not detected in the WWTP, thus offering a balance between WWTPs and clinical sequencing.

It is important to track and characterize mutations at large volume facilities as well as obtain a higher resolution at a local level, which is much closer to the potential source. Such methods when employed systematically can help in deploying mitigating efforts to contain outbreaks. Surveillance, quantification, sequencing, and comparison of results is intricate as wastewater is a complex matrix. Additionally, the impact of temporal and geographical variability needs to be carefully assessed for data from wastewater surveillance to be meaningful. Therefore, this comparison study is valuable and important. However, there are concerns regarding the methodology and the conclusions derived from the results, which need to be addressed before the paper can be published.

Major issues

1- Lines 227-228. To compare the read frequencies differences between the WWTP influent collected in July and August, the WWTP influent collected in September, and the university wastewater samples collected in September, the authors use the student’s t-test. The t-test is generally used for pairwise comparisons, for comparing three or more conditions, ANOVA for multiple comparisons is more suitable. The authors should use ANOVA and assess the significance of the differences detected.

2- Another limitation that should be addressed is that at low RNA copy numbers, the read frequencies cannot be correlated as absolute frequencies of mutations found in a sample. The aliquots used for cDNA synthesis will show stochastic variations due to the low RNA copy numbers. This results in biased PCR amplifications and the read frequencies might vary among the replicates from the same sample. Therefore, at low copy numbers (as seen in the Columbia WWTP), comparing read frequencies may not be appropriate.

3- These caveats are also problematic as read frequencies were used for the probabilistic models in the co-occurrence and linkage analysis. The methodologies used for the co-occurrence and linkage analysis should be further detailed in the paper.

4- The use of the terms co-occurrent and concurrent should be explained in more detail. For example, in line 293, “here” implies that in other instances co-occurring has another meaning? Additionally, the authors should explain why none of the pairs of mutations identified by linkage overlapped with those identified through the R package cooccur.

Minor issues

1- More precise information would be useful for Figure 1. What were the cases of isolation or quarantine based on? Did they test positive for SARS-CoV-2 or as suggested by the legend suspected of exposure?

2- Line 170- Figure 2 is mentioned to visualize site 1 but Figure 4, panel B should be referenced to show the overlapping points. The authors use the PCA analysis showing overlapping points for the UofSC site 1 for August 28 and September 11 to suggest sequence stability. The average shedding time of the SARS-CoV-2 virus is two weeks, the same time lapse between the two sampling events at site 1. Therefore, the authors should explain the sequence stability because of the temporal and spatial distribution of the samples.

3- Figure 4. The color coding of the groups has been switched on panel A. Red should be assigned to Sept and blue to July/Aug.

4- Line 287. Something seems to be missing. “complementary techniques to identify co-occurrence. patterns in sequence space.”

Reviewer #2: In the submitted manuscript “Wastewater surveillance illustrates overlapping SARS-CoV-2 mutational profiles between a university campus and its surrounding community”, the authors present the results of municipal-level wastewater treatment plant (WWTP) surveillance and campus-level surveillance in concert. They identify common trends and also differences between the WWTP and campus samples along with presentation of results of a variety of statistical tools that aim to deconvolute the wastewater mixtures into co-occurring genotypes. I have a few concerns and suggestions listed below:

1) Interpretation of the SNPs over time (Page 10-12): The co-occurrence result is challenging to interpret, especially the linkage results on page 13. I like Figure 6 A and B, and think they should stay in some form (and maybe even largely as-is), but they need more interpretation. What viral context do these networks reveal?

The authors state on page 10 (lines 209-221) that Cluster 4 mutations, including 25563 g>t, are found in all samples at close to 100% frequency, which would suggest that at the time(s) of sampling, nearly all detectable SARS-CoV-2 in Columbia and UofSC are of the 20C clade (or a descendant). However, one of the other major markers for clade 20C, 1059 c>t, is found in fewer enough samples that it is grouped into the “all conditions, albeit less consistently” Cluster 3. To my eye, the differences between 1059 and 25563 are not huge in the campus samples. All but one (site 1, 9/11) have both. From this we might surmise a strong presence of 20C (or descendant) on the campus at the time of sampling. The less consistent signal in the WWTP samples suggest a more complicated mix of genotypes in the broader area.

Alternatively, if we expect to find 1059 in all samples where the 25563 mutant is found, might that imperfect correspondence be the result of wastewater sequencing itself, with this more challenging substrate producing more piecemeal results as one of the sensitivity tradeoffs? Along these same lines of sensitivity, I note that there is no discussion of the spike mutation D614G (23403 a>g). Do the V3 amplicons poorly amplify this region? Can the authors speculate on the reasons?

2) A 20G outbreak on campus: I think at least one of the things the authors have revealed (Networks in Fig 6, SNPs in Fig 5/ S1, and lists thereof on pages 10-12), is it looks like they caught a 20G outbreak in-progress on campus. Hints of which are seen in the WWTP, but not nearly as consistently as the 20G signal caught in-the-act on campus. The authors should emphasize this finding/interpretation, even as it may require re-writing of the co-occurrence results.

On lines 230-232, the authors describe a significance test whereby 6 mutations were different between September WWTP and University samples “1358g > a, 3037c > t 13,201g > t, 18,424a > g, 25,907g > t, and 27,964c > t.” Of these, 3037 c >t is hard to explain (it is found in all 20A and all descendants, along with 241, 14408, and 23403), but 18424, 25907, and 27964 are some of the markers of 20G. If these 3 are all going in the same direction (are they?), they may be evidence that either the University or the City were undergoing a wave of 20G that the other was not (https://github.com/nextstrain/ncov/blob/master/defaults/clades.tsv).

The tests continue on lines 232-236, with summer City samples compared to University samples. Two more 20G markers make this list: 10319 and 28472. Another universal 20A marker does too (241 c>t). I think the interpretations of each are similar to those on the previous tests.

Together, the above, and knowing the global context of mutations 10319, 18424, 25907, 27964, and 28472 as markers of 20G, the authors may have bona-fide evidence of a 20G outbreak in-progress on this campus. While less consistent hints of 20G are found in the City samples, the 20G signal is quite “bright” in 3 of the campus samples (Fig S1), (Site 1 8/28, Site 8 8/28, Site 5 9/4) and absent in (Site 11 9/4, Site 1 9/11).

3) PCA plots in 4B: The PCA in Figure 4B does not seem to agree with the other findings of sample similarity/dissimilarity. The two samples in the center are unlikely to be that similar.

I find it hard to come up with a scenario in which two samples would completely overlap on a PCA plot unless they were identical (UofSC points 1 8/28/20 and 1 9/11/20). I guess it could be possible if their differences only showed up on an N-th PC and not PC1 or 2? At minimum, the authors should check their input matrix/dataframe for the PCA in 4B, verify whether these two datapoints are identical or not, and add/expand comments on this to help explain. Figure 4’s legend needs to also explain what is being plotted, not just “samples”. Is it a singular inferred genotype per sample or a species proportion?

Moreover, regardless of what interpretation of genotype(s) is being computed for a dissimilarity score, it is difficult to believe that these two samples in particular would be perfectly identical on the first two PCs, given how many SNPs they have not-in-common, per figure 5 Clusters 3,2, and 1.

Other issues

Introduction

The need to have Figure 5 and Figure S1 which only differ by the S1 having text labels on each row is odd? Figure S1 seems to be sufficient, and an idea would be to merge the two figures back into one, and in the body of the article.

Line 42-43: The statement that “reinfection with SARS-CoV-2 has been demonstrated to occur on college campuses in at least 16% of students” is interpreted incorrectly from Reference #8. The estimated reinfection rate in Reference 8 was 2.2% with 16% of the population susceptible to reinfection. This needs to be fixed.

Figure 1

Figure 1’s X axis makes sense in set notation, but probably not as what most readers will expect from a scientific figure. Would it be appropriate to replace the current x-axis label with the latter date of each given set’s range?

Figure 3

Fig 3A is challenging to interpret. The gray dots along the axes aren’t easy to mentally place alongside the colorful diagonal dots. I advise the authors to rework this figure to 3 horizontal graphs, such as a top level with the gray UofSC-only sites, a middle with the colorful shared sites, and a bottom level with Columbia WWTP-only sites. Dot jitter could be used for legibility. Also, the Y-axis on Fig 3A uses the abbreviation “USC” not found elsewhere in the manuscript.

Figure 4

Axis labels and legends lack capitalization. Name the 4A legend something like “WWTP groups.”

Linkage results and discussion

The linkage discussion on divergence on p 13 is incomplete and could be made more clear. What is the significance or lack thereof of the disagreements? Also, there are some typos on line 293.

Other minor notes

On line 78, MGD has not been defined.

On and around line 100, which native barcoding kit(s) were used? Expansion 1-12, 13-24, Expansion 96?

On line 104, unless all amplicons are exactly 400 bp, replace with “~400 bp.”

On line 113, was the flow cell R9.4? If possible, more specific is better.

Citation 16 only has author initials.

6. PLOS authors have the option to publish the peer review history of their article (what does this mean?). If published, this will include your full peer review and any attached files.

Reviewer #1: No

Reviewer #2: **Yes: **Edwin Oh, Van Vo, and Richard Tillett

---

## [Author Response · Author response to Decision Letter 0]

1 Feb 2022

Please see the uploaded file labeled "Response to Reviewers"

---

## [Decision Letter · Decision Letter 1]

21 Mar 2022

Wastewater surveillance of SARS-CoV-2 mutational profiles at a university and its surrounding community reveals a 20G outbreak on campus

PONE-D-21-32875R1

Dear Dr. Norman,

We’re pleased to inform you that your manuscript has been judged scientifically suitable for publication and will be formally accepted for publication once it meets all outstanding technical requirements.

Kind regards,

Theodore Raymond Muth

Academic Editor

PLOS ONE

Additional Editor Comments (optional):

Reviewers' comments:

Reviewer's Responses to Questions

**Comments to the Author**

1. If the authors have adequately addressed your comments raised in a previous round of review and you feel that this manuscript is now acceptable for publication, you may indicate that here to bypass the “Comments to the Author” section, enter your conflict of interest statement in the “Confidential to Editor” section, and submit your "Accept" recommendation.

Reviewer #1: All comments have been addressed

Reviewer #2: All comments have been addressed

2. Is the manuscript technically sound, and do the data support the conclusions?

Reviewer #1: (No Response)

Reviewer #2: Yes

3. Has the statistical analysis been performed appropriately and rigorously? 

Reviewer #1: (No Response)

Reviewer #2: Yes

4. Have the authors made all data underlying the findings in their manuscript fully available?

Reviewer #1: (No Response)

Reviewer #2: Yes

5. Is the manuscript presented in an intelligible fashion and written in standard English?

Reviewer #1: (No Response)

Reviewer #2: Yes

6. Review Comments to the Author

Reviewer #1: (No Response)

Reviewer #2: I commend the authors for a job well done. This is an interesting finding and all of our concerns have been addressed.

7. PLOS authors have the option to publish the peer review history of their article (what does this mean?). If published, this will include your full peer review and any attached files.

Reviewer #1: No

Reviewer #2: **Yes: **Edwin Oh

---

## [Editor Report · Acceptance letter]

4 Apr 2022

PONE-D-21-32875R1 

Wastewater surveillance of SARS-CoV-2 mutational profiles at a university and its surrounding community reveals a 20G outbreak on campus 

Dear Dr. Norman:

I'm pleased to inform you that your manuscript has been deemed suitable for publication in PLOS ONE. Congratulations! Your manuscript is now with our production department. 

Kind regards, 

on behalf of

Dr. Theodore Raymond Muth 

Academic Editor

PLOS ONE